# Attract the best: The attraction effect as an effective strategy to enhance healthy choices

**Gitta van den Enden** * , **Kelly Geyskens**

Department of Marketing and Supply Chain Management, Maastricht University, Maastricht, The Netherlands

These authors contributed equally to this work.
* g.vandenenden@maastrichtuniversity.nl

## Abstract

Every day, people make many food decisions without thinking, repeatedly falling for the unhealthy option instead of the healthy option. While making these mindless decisions, people often rely on heuristics. In this paper, we demonstrate that these heuristics can be exploited to nudge consumers towards healthy alternatives. Specifically, we explore how the attraction effect (i.e., adding a decoy to a choice set) can nudge people to choose a healthy snack. The results of our choice experiment indicate that adding a decoy (i.e., a less attractive food alternative) to a self-control situation (i.e., choosing between a healthy and an unhealthy food alternative) can help people maintain self-control and choose the healthy option. This mixed choice set thus nudges people towards the healthy option. Moreover, our results show differential effects of the attraction effect depending on the (un)healthiness of the products in the choice set. Specifically, the attraction effect is prominent when the choice set consists of unhealthy products only (i.e., the unhealthy choice set), but not in the choice set that consists of only healthy products (i.e., healthy choice set). Importantly, our results indicate when the attraction effect can exploit consumers' heuristics to help them make better, healthier food choices.

## 1. Introduction

Despite their good intentions to eat healthier, people, without thinking, often choose unhealthy food instead of healthy options as a result of impulsive tendencies [1]. That is, food decisions are often made impulsively, overriding people's long-term goal of eating healthy. This is true because people make many decisions each day and therefore often lack the resources to make rational, well-considered decisions [2, 3]. These limited cognitive resources make people more impulsive and more prone to salient cues or heuristics [4–6]. Interestingly, this offers opportunities for heuristics to be exploited to trigger healthy choices by making the healthy choice dominating.

The attraction effect is one of the most well-known and effective strategies to make a particular choice option dominating, and hence, offers a potentially viable route to trigger healthy choices. This frequently investigated context effect has an influence on the purchase behavior of consumers, especially when people's choices are based on heuristics [7]. The attraction effect

**Funding:** The author(s) received no specific funding for this work.

**Competing interests:** The authors have declared that no competing interests exist.

implies that adding an asymmetric option (i.e., the decoy) to an existing choice set (consisting of a target and a competitor product) that is unmistakably dominated by the target product, but not by the competitor, can increase the choice share of the asymmetrical dominating target product. It is found to be robust in many hypothetical choice contexts and has been investigated with dozens of different products (e.g., microwaves, running shoes, computers, and beer) and in many diverse choice contexts (e.g., deciding to screen for colorectal cancer [8] and choosing a job candidate [9] or a political candidate [10]). In fact, the attraction effect changes the environment, which might be a more effective strategy than trying to convince people what is "right" [11].

However, the extent to which consumers base their decisions on heuristics is not the same in every choice context. For example, consumers who are choosing between unhealthy products—which is a hedonic type of food product providing gratification from sensory attributes like taste [12]—are more likely to use their sensory system and heuristics compared to choosing between healthy alternatives—which is a utilitarian food type of food product focusing on instrumental attributes like ingredients—in which functionality and instrumental motivations are mostly used [13]. Related, recent research has shown that context effects work differently for hedonic and utilitarian products. More specifically, the compromise effect (i.e., people tend to prefer options positioned as a compromise in a given set of extreme options [7]) is strongly elicited in choice sets consisting of utilitarian products, whereas the effect does not occur in choice sets consisting of hedonic products [14]. With utilitarian choice sets, consumers make decisions based on value calculations, which increases the chance of consumers choosing a compromise option. Conversely, research shows that the attraction effect is more effective when people use heuristics. More specifically, promotion-focused (vs. prevention-focused) consumers are more triggered to capture that one-time opportunity and are therefore more likely to use heuristics, which makes them more susceptible to the attraction effect [15]. Uniting literature demonstrating that people are more likely to make decisions based on heuristic for hedonic (vs. utilitarian) products with the findings that the attraction effect is more effective when people use heuristics, we posit that the attraction effect is stronger in hedonic choice sets than with utilitarian choice sets.

Importantly, besides examining the attraction effect in hedonic and utilitarian choice sets, we investigate the effect in a mixed choice set (i.e., a choice set consisting of both hedonic and utilitarian product(s)) reflecting a typical self-control situation (i.e., choosing between a healthy and an unhealthy food alternative). Importantly, in such situations that require a certain amount of self-control, people often use heuristics instead of value calculations and cognitive strategies because of limited cognitive capacities [2, 3]. Since the attraction effect is particularly effective when people make use of heuristics, we argue that we can use these heuristics as endorsed by the attraction effect to help people maintain self-control. In particular, the attraction effect might be especially useful to nudge people towards the utilitarian option that is in line with long-term goals. For example, when having to choose between an apple (i.e., the utilitarian option) and a muffin (i.e., the hedonic option), the addition of a third (decoy) option might enable you to override your impulse to choose the muffin. Specifically, the addition of a third (decoy) option which is dominated by the apple makes the apple the dominating option in the choice set. Because you rely on heuristics, this choice set -illustrative of the attraction effect- will actually lead you to choose this clearly dominating healthy option. Therefore, in this paper we hypothesize that the addition of a decoy to a choice set consisting of a utilitarian target option and a hedonic competitor option, whereby the utilitarian target dominates the added decoy, will increase the choice share of this utilitarian target option.

In sum, the present study hypothesizes that (i) the attraction effect is present in unhealthy choice sets (i.e., choice sets consisting of only unhealthy products), but not in healthy choice

sets (i.e., choice sets consisting of only healthy products), and that (ii) the attraction effect can be used as a nudge toward healthy food choices (i.e., prefer a healthy to an unhealthy food option). Investigating whether adding a decoy to the choice set can help people adhere to their long-term goals and make better choices is extremely relevant and important in the light of the current obesity problem.

## 2. Method

We created three choice sets: one consisting of only unhealthy products, one with only healthy products and one mixed choice set consisting of a healthy target, an unhealthy competitor and a healthy decoy product. To the best of our knowledge, previous research on the attraction effect has studied the attraction effect using choice sets that consisted of three similar products (e.g., three video cameras). In those studies, the products in the choice set differed only in specific product characteristics (e.g., price, quality, number of features). However, in many food choice situations, consumers do not choose between three similar products, (e.g., several orange juices that differ on price and quality rating), but rather between products that differ on several characteristics (e.g., orange juice, milk, or water). Therefore, in our research we aim for a more realistic setting by creating choice sets that are more likely to fit in common consumption situations. More specifically, all products in our choice sets are small snacks that people can pick if they are looking for a quick bite to eat.

### 2.1. Pre-test

When testing the attraction effect, the choice sets should consist of two products that are equally attractive (target and competitor) and one product that is less attractive and dominated by the target (decoy) [7]. In a separate study prior to the main study, we pretested different products with respect to their perceived attractiveness level to ensure similarity on this respect. Furthermore, to be able to make a distinction between unhealthy and healthy products, we also measured the perceived level of healthiness of each product. Participants rated 25 products (e.g., snack tomatoes, snack cucumbers, grapes, raisins, wine gums, sweets, and chocolate waffles) in randomized order on their level of healthiness and attractiveness, both on a scale 0–100 (N = 25). Based on these scores, we compiled the different choice sets in such a way that two products per choice set are equally attractive (i.e., the target and competitor) and one is significantly less attractive (i.e., the decoy). Table 1 presents the selected products. A complete overview of their mean level of attractiveness and healthiness can be found in S1 Table. In all choice sets, the level of attractiveness of the selected target and competitor product are not significantly different, whereas the attractiveness of the decoy is significantly lower than both the target and competitor. In the unhealthy choice set, the products do not differ on their level of unhealthiness and in the healthy choice set all three products are perceived as a healthy snack. Lastly, the competitor in the mixed choice set is rated as significantly less healthy than both the target and decoy.

### 2.2. Main study

**2.2.1. Participants and study design.** A sample of 237 participants (76.4% female, $M_{age}$ = 35.5) were recruited via convenience sampling and completed the online survey, without any guaranteed compensation for the participants beforehand. However, to encourage the participants to participate and to increase the realism of the study, the respondents had the chance to win one of their chosen products. We applied a 2 (attraction effect manipulation: no-decoy or decoy) × 3 (type of choice set: unhealthy, healthy and mixed) mixed design. Attraction effect was a between factor, and type of choice set was a within factor. In particular, the participants

**Table 1. Choice sets main experiment.**

| Choice set | Target | Competitor | Decoy |
|---|---|---|---|
| Unhealthy choice set | M&M's | Bonbons | Sweets |
| Healthy choice set | Snack Tomatoes | Unsalted Cashews | Granola Cookies |
| Mixed choice set | White Grapes | Chocolate Chip Cookies | Carrots |

were randomly assigned to the no-decoy (i.e., 2 products to choose from: the competitor and the target) or the decoy condition (i.e., 3 products to choose from: the competitor, the target, and the decoy). Moreover, all participants made a choice from the three choice sets (unhealthy, healthy and mixed) in randomized order. We can draw the same statistical conclusions when we only include the first choice set participants saw in the analyses. Also the products within each choice set were displayed in randomized order to control for natural preference for the middle option [16]. We received ethical approval from the university's ethics committee (Behavioral & Experimental Economic Laboratory (BEELab)) (approval number 18006). Furthermore, the authors ensure that the work described has been carried out in accordance with The Code of Ethics of the World Medical Association (Declaration of Helsinki) for experiments involving human subjects. Written informed consent was obtained at the beginning of the experiment by all participants. Additionally, participants were given the option to receive a debriefing. In all stages of the experiment, the privacy rights of human subjects were observed.

**2.2.2. Procedure.** Approximately half of the participants were presented with only the competitor and the target product in their three choice sets (no-decoy condition). The other participants had to choose their preferred product out of the target, the competitor and the decoy (decoy condition). In this way, it is possible to determine whether the target is chosen more often in the decoy condition than in the no-decoy condition, and hence whether the choice sets exhibit the attraction effect.

To construct the choice sets in line with the attraction effect theory (in which the target asymmetrically dominates the decoy, but not the competitor), we used two dimensions to describe the products; "taste rating" and "quality of ingredients". We constructed the choice sets with the values of the product dimensions to make them as coherent as possible with the product categories. More specifically, the dimension "taste rating" is a more hedonic characteristic, whereas "quality of ingredients" has a more utilitarian nature. Therefore, in the unhealthy choice set, the target dominates the decoy on the dimension taste rating, while, in the healthy and mixed choice set, the healthy target product dominates the decoy on the dimension quality of ingredients. An overview of the choice sets can be found in S2 Table.

Furthermore, we included several measures into the questionnaire that allow us to control for in the analyses. First, to measure the level of self-control, we included a 13-item Brief Self-Control Scale [17] in the survey (e.g., "I am good at resisting temptations" and "I have trouble concentrating"). Participants indicated on a 5-point scale ranging from "*not at all*" to "*very much*" how well the statements reflect themselves. Analyses revealed that the scale had a high level of internal consistency ($\alpha = 0.80$). Respondents were also asked to indicate whether they were currently on a diet and specified their gender and age in the survey.

## 3. Results

Table 2 gives an overview of the share of the target product relative to the competitor product. In our analyses, we do not take the share of the decoy product into account, because the difference of number of products in the two conditions (2 products in the no-decoy vs. 3 products

**Table 2. Relative share of the target product per condition.**

| Choice set | Share target in no-decoy condition | Share target in decoy condition |
|---|---|---|
| Unhealthy choice set | 52.5% | 76.0% |
| Healthy choice set | 53.4% | 42.0% |
| Mixed choice set | 54.2% | 72.7% |

in the decoy condition) could bias our results [18]. Hence, we only consider the share of the target and competitor product, which are present in both conditions.

To test our hypotheses, we ran two types of statistical analyses. First, we ran binary logistic regressions in each choice set to test whether the choice share of the target option in the decoy condition differs from the share in the no-decoy condition. Self-control, diet, gender and age are added as control variables. Second, we run binomial tests to investigate whether the choice shares of the target and competitor product differ from 50%, and hence whether participants had a clear preference for one of the two products or were indifferent.

A binary logistic regression in the choice set consisting of only unhealthy products with self-control, diet, gender and age as control variables reveals a significant attraction effect. More specifically, the relative share of the target is higher in the decoy condition (76.0%) than in the no-decoy condition (52.5%; ß = 1.113, $\chi^2$ (1) = 11.85, $p$ = .001, Nagelkerke $R^2$ = .216). Importantly, a binomial test reveals that whereas the target share in the no-decoy condition (52.5%) is not significantly different from 50% ($p$ = .645), the preference for the target is significantly higher than 50% in the decoy condition (76%, $p$ = .000). In contrast, the findings in the healthy choice set reveal that the target share of the decoy condition (42.0%) is marginally significantly lower than in the no-decoy condition (53.4%; ß = -.543, $\chi^2$ (1) = 3.79, $p$ = .052, Nagelkerke $R^2$ = .103). Importantly, however, the binomial test indicates that the target share does not differ from 50%, neither in the no-decoy condition (53.4%, $p$ = .519) nor in the decoy condition (42.0%, $p$ = .108), which implies that there is no strong preference for either the target or the competitor product in the two conditions in the healthy choice set. Hence, as hypothesized, the choice set consisting of unhealthy products does elicit a significant attraction effect, but the healthy choice set does not.

Interestingly, as with the unhealthy choice set, the relative shares in Table 2 indicate that the mixed choice set produces an attraction effect as well. Indeed, consistent with our prediction, the relative choice share of the target increases when we add the decoy to the choice set (54.2% vs. 72.7%; ß = .876, $\chi^2$ (1) = 8.84, $p$ = .003, Nagelkerke $R^2$ = .118). Likewise, the binomial test shows that while there is no preference in the no-decoy condition for one of the products (share of the target is 54.5%, $p$ = .407), there is a clear preference for the target product in the decoy condition (72.7%, $p$ = .000). Thus, adding a healthy decoy to a choice set consisting of a healthy target and an unhealthy competitor product makes consumers choose the healthy target over the unhealthy competitor product.

## 4. Discussion and conclusion

In line with our predictions, our results indicate that the attraction effect is not equally effective for every food product category. We found a significant attraction effect in a choice set consisting of three unhealthy products, whereas the attraction effect is rather absent in the healthy choice set. While the target share of the decoy condition tends to be lower than in the no-decoy condition in the healthy choice set, people remained indifferent between the target and competitor in both the no-decoy and the decoy condition. These results suggest that the attraction effect seems to be an effective nudging strategy for hedonic choice sets, but not for utilitarian choice sets.

Moreover, we also find a significant attraction effect in the mixed choice set. Importantly, this supports our prediction that adding a decoy enables consumers to make healthier food choices. As has been argued in previous research [11], it is better to adjust the environment so that consumers are nudged towards a healthy product instead of explicitly convincing consumers to choose a healthy alternative instead of an unhealthy one. This study adds the attraction effect as another successful method of nudging consumers towards healthy products.

One alternative explanation for the effect found in the mixed choice set could be increased salience of healthy products. In particular, one could argue that the choice share of the healthy target in our mixed choice set increases with the addition of the healthy decoy, because there are two healthy options present in the choice set and one unhealthy option. To exclude this alternative explanation, we conducted a small follow-up study that added an unhealthy decoy. This way we can determine whether it is the healthiness of the added decoy or rather the dominance of the target product over the added decoy, that drives the attraction effect found in the mixed choice set. We tested the effect with an unhealthy decoy (hard candy) added to a choice set consisting of a healthy target (blueberries) and an unhealthy competitor product (bonbons). Interestingly, the attraction effect also prevailed in this choice set. This implies that we can help consumers choosing the healthy option over the unhealthy option by adding a decoy, regardless of its level of un/healthiness.

The findings of our study offer several practical implications. First, the results of our mixed choice set show that when practitioners or policy makers want to nudge consumers towards healthy food products, they should add a decoy to the choice set that is being dominated by the healthy target product. This offers new opportunities for assisting consumers in choosing healthy products over unhealthy products, which is extremely important considering the worldwide overweight and obesity problem [19] and the ongoing Covid-19 pandemic that has led to a decrease in healthy food intake [20], especially among overweight and obese individuals [21]. Second, we found that this strategy does not apply to all kinds of choice sets. More specifically, when practitioners construct a choice set that consists of only unhealthy options, the target option should -as in the mixed choice set–dominate the decoy product. Moreover, the attraction effect might not be the best strategy for choice sets that consist of only healthy products and practitioners could alternatively consider to make use of the compromise effect [14].

Although the results of this study reveal several important findings, we look forward to see whether the findings extend beyond a hypothetical, online context. For example, it would be of practical relevance to investigate whether our findings hold (i) in more realistic settings such as a supermarket, (ii) when using more than two dimensions to describe the products and (iii) when including products from other categories than food. Moreover, as our sample mainly consists of females (76.4%), our results might be biased towards this group. Previous research has shown that—compared to males—females are more likely to try to lose weight and eat healthy [22]. Although neither the interaction of condition (decoy vs. no-decoy) with gender, nor with diet on target choice share are significant in our study, we encourage further research to shed light on gender differences in the case of the attraction effect.

The results in this study suggest that the attraction effect is a method of nudging consumers towards better, healthier choices, which encourages healthy behavior of consumers and contributes to solve the overweight problem. Importantly, using a decoy to increase the likelihood that consumers choose the dominating target option is especially effective in situations that require a certain amount of self-control (i.e., unhealthy and mixed choice sets). These findings suggest that the attraction effect can be an effective strategy to help people maintain self-control.

## Supporting information

**S1 Table. Choice sets with mean perceived level of attractiveness and healthiness.**
(PDF)

**S2 Table. Choice sets main experiment.**
(PDF)

## Author Contributions

**Conceptualization:** Gitta van den Enden, Kelly Geyskens.

**Formal analysis:** Gitta van den Enden.

**Investigation:** Gitta van den Enden, Kelly Geyskens.

**Methodology:** Gitta van den Enden, Kelly Geyskens.

**Project administration:** Gitta van den Enden.

**Supervision:** Kelly Geyskens.

**Validation:** Gitta van den Enden, Kelly Geyskens.

**Visualization:** Gitta van den Enden.

**Writing – original draft:** Gitta van den Enden, Kelly Geyskens.

**Writing – review & editing:** Gitta van den Enden, Kelly Geyskens.

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
