## [Decision Letter · Decision Letter 0]

10 Aug 2021

PONE-D-21-15713

Attract the best: The attraction effect as an effective strategy to enhance healthy choices

PLOS ONE

Dear Dr. van den Enden,

Thank you for submitting your manuscript to PLOS ONE and for your patience. After careful consideration, and based on the feedback offered by two expert reviewers alongside my reading of the paper, I feel it has merit. Still, it would benefit from minor revision to fully meet PLOS ONE’s publication criteria. Therefore, I invite you to submit a revised version of the manuscript that addresses the points raised during the review process, which qualify, as stated earlier, for a minor revision.

We look forward to receiving your revised manuscript.

Kind regards,

Ali B. Mahmoud, Ph.D.

Academic Editor

PLOS ONE

Journal Requirements:

2. Please specify type of consent obtained.

Additional Editor Comments:

Reviewers' comments:

Reviewer's Responses to Questions

**Comments to the Author**

1. Is the manuscript technically sound, and do the data support the conclusions?

Reviewer #1: Partly

Reviewer #2: Yes

2. Has the statistical analysis been performed appropriately and rigorously? 

Reviewer #1: Yes

Reviewer #2: N/A

3. Have the authors made all data underlying the findings in their manuscript fully available?

Reviewer #1: Yes

Reviewer #2: Yes

4. Is the manuscript presented in an intelligible fashion and written in standard English?

Reviewer #1: Yes

Reviewer #2: Yes

5. Review Comments to the Author

Reviewer #1: This manuscript presents a pre-test and the main study investigating the effect of the attraction effect, as a nudging strategy, on the choice of healthy snacks. Results show that the attraction effect is stronger in choice sets with exclusively unhealthy options than in choice sets with exclusively healthy options. The mixed condition shows that healthy options can be preferred over unhealthy if presented as target options in the choice set. The manuscript is well written, and the story that the authors present is simple and clear. While I think the manuscript makes a valuable contribution to the literature, some details could improve it. Below the authors find my main comments:

1. The association between levels of healthiness and hedonic and utilitarian characteristics of foods is not clear in its theoretical explanation in the introduction (lines 51-55). It is novel, for my knowledge, and not introduced neither in Dahr and Wertenbroch (2000) or in Kim and Kim (2016), a demonstrated association between hedonic or utilitarian and level of the healthiness of food. It is not easy to understand from the introduction the reasons behind this association. It became more apparent when in the methodology, the authors use "taste rating" and "quality of ingredients" as attributes of the snacks (line 148). However, it might be a good idea either, if the authors measured it, to report a direct assessment of how much participants perceived each option as hedonic and/or utilitarian and specify better the association in the introduction.

2. The study's aims are interesting; however, it is more common to formulate specific hypotheses in this kind of literature. I think the formulation of specific hypotheses will make the goals of the study clearer.

3. The methodology is very well explained and complete. However, there are some details that I think can improve the paper. Firstly, the pre-test sample is not specified; is it the same as the main study? Was the pre-test done in the same session of the main study or before? Secondly, it is unclear who the main study sample is; are they students, or were the data collected through an online platform? If the authors used an online platform (e.g., Mturk or Prolific), how did you assure people receiving one of the products?

4. The results are interesting and well presented. I have only a few comments. What is the statistic that the authors used to compare the strength of the attraction effect in healthy product conditions and unhealthy product conditions (line 189)? The authors can conclude that one is significant and the other just marginal; however, it is not clear how the authors can conclude that one is stronger than the other one. Moreover, when performing a binary logistic regression is more complete to report also the beta coefficients and the R2, while in the analysis there is only a chi-square. It is also not clear if the authors reported there (e.g., line 181) the regression results where they controlled for self-control, diet, gender, and age or the one without those variables. Maybe the regression table could also be reported in the results section since the paper is not too long.

5. Finally the discussion is well written and well motivated.

In general, I think the manuscript makes a valuable contribution, it is well written, and if the few reported details were addressed, I would suggest acceptance.

Reviewer #2: I though this paper is well written, and information is presented in a concise and relevant way, which I appreciate. The novelty of this study comes from the novel experiment design which allows exploring effects of attractions across new product mix sets.

The results are interesting from the practical point of view, although this is just a first step as validating the findings beyond an online experiment in a practical setting and for various product types is needed. Since this study is like a gateway towards more realistic and more product inclusive further studies, I believe the concluding section could benefit from more extensive coverage of what needs to be done in the future and where these types of results may have best use. I think that my suggestion in 1) below should take care of this.

I would suggest few revisions to improve the overall context and the flow to the reader.

1) I think the limitations of this study, which are summarized in one short paragraph (lines 238-241) should be expanded and communicated more formally. I would think that participant’s demographics may have an impact on the results of this experiment. For example, current sample was mostly females, perhaps the results are biased towards this group, thus gender, as well as generational attributes (age of the participants) may also be contributing factors in the results found in this study.

2) It may be wise to include in the discussion of an ongoing Covid-19 Pandemic, as it may be also proving to have severe negative emotional impact on people’s eating habits (well documented in research studies by now). Depending on extend of the relevance to the current study, this could be accomplished in the introduction section or discussion. The study does mention repeatedly the importance of the environment, thus pandemic impact may be linked to that.

3) I would like to see illustrative example with an apple and muffin (line 77) to be complete with an example of a decoy product choice appropriate for such a product mix. It reads unfinished.

4) Line 185: change significant to significantly

Several references might be useful here:

Brody, L. R., Hall, J. A. & Stokes, L. R. (2018). Gender and emotion: Theory, findings, and context. In L. F. Barrett, M. Lewis & J. M. Haviland-Jones (Eds.), Handbook of emotions (4th ed., pp. 369– 392).

Bove, C. F., Sobal, J. & Rauschenbach, B. S. (2003). Food choices among newly married couples: Convergence, conflict, individualism, and projects. Appetite, 40, 25– 41.

Marty, L., de Lauzon-Guillain, B., Labesse, M. & Nicklaus, S. (2021). Food choice motives and the nutritional quality of diet during the COVID-19 lockdown in France. Appetite, 157, 105005. https://doi.org/10.1016/j.appet.2020.105005

Nazzaro, C., Lerro, M. & Marotta, G. (2018). Assessing parental traits affecting children’s food habits: An analysis of the determinants of responsible consumption. Agricultural and Food Economics, 6, 23.

Poelman, M. P., Gillebaart, M., Schlinkert, C., Dijkstra, S. C., Derksen, E., Mensink, F. et al. (2021). Eating behavior and food purchases during the COVID-19 lockdown: A cross-sectional study among adults in the Netherlands. Appetite, 157, 105002. https://doi.org/10.1016/j.appet.2020.105002.

Mahmoud, A. B., Hack-Polay, D., Fuxman, L., & Nicoletti, M. (2021). The Janus-faced effects of COVID-19 perceptions on family healthy eating behavior: Parent’s negative experience as a mediator and gender as a moderator. Scandinavian journal of psychology. https://doi.org/10.1111/SJOP.12742

6. PLOS authors have the option to publish the peer review history of their article (what does this mean?). If published, this will include your full peer review and any attached files.

Reviewer #1: No

Reviewer #2: No

---

## [Author Response · Author response to Decision Letter 0]

23 Sep 2021

Attract the best: The attraction effect as an effective strategy to enhance healthy choices.

(PONE-D-21-15713)

Response to reviewers

Thank you very much for offering us the opportunity to revise our work. We were encouraged that you find our topic – the attraction effect as an effective strategy to enhance healthy choices – interesting and important. We addressed the issues raised by the review team. Below, we repeat all comments in italics and then respond to all comments. We believe this revision substantially improved the paper. 

Journal Requirements:

We made minor adjustments to meet PLOS ONE’s style requirements (e.g., file names and titles of Supporting Information)

2. Please specify type of consent obtained.

We obtained written informed consent at the beginning of the experiment by all participants. We added this statement in 2.2.1.

We included our ethics statement including the name of the ethics committee and type of consent in 2.2.1. 

We included the captions for the Supporting Information files at the end of our manuscript. Thank you for your comments. 

Additional Editor Comments:

We reviewed the reference list and made minor adjustments to correct details (i.e., finetuning of page numbers and DOI use).

Reviewers' comments:

Reviewer's Responses to Questions

Comments to the Author

1. Is the manuscript technically sound, and do the data support the conclusions?

Reviewer #1: Partly

Reviewer #2: Yes

2. Has the statistical analysis been performed appropriately and rigorously? 

Reviewer #1: Yes

Reviewer #2: N/A

3. Have the authors made all data underlying the findings in their manuscript fully available?

Reviewer #1: Yes

Reviewer #2: Yes

4. Is the manuscript presented in an intelligible fashion and written in standard English?

Reviewer #1: Yes

Reviewer #2: Yes

5. Review Comments to the Author

Reviewer #1: This manuscript presents a pre-test and the main study investigating the effect of the attraction effect, as a nudging strategy, on the choice of healthy snacks. Results show that the attraction effect is stronger in choice sets with exclusively unhealthy options than in choice sets with exclusively healthy options. The mixed condition shows that healthy options can be preferred over unhealthy if presented as target options in the choice set. The manuscript is well written, and the story that the authors present is simple and clear. While I think the manuscript makes a valuable contribution to the literature, some details could improve it. Below the authors find my main comments:

1. The association between levels of healthiness and hedonic and utilitarian characteristics of foods is not clear in its theoretical explanation in the introduction (lines 51-55). It is novel, for my knowledge, and not introduced neither in Dahr and Wertenbroch (2000) or in Kim and Kim (2016), a demonstrated association between hedonic or utilitarian and level of the healthiness of food. It is not easy to understand from the introduction the reasons behind this association. It became more apparent when in the methodology, the authors use "taste rating" and "quality of ingredients" as attributes of the snacks (line 148). However, it might be a good idea either, if the authors measured it, to report a direct assessment of how much participants perceived each option as hedonic and/or utilitarian and specify better the association in the introduction.

We agree there was room for clarification on this part, thank you for pointing this out. We rephrased lines 51-55 to clarify this early on.

2. The study's aims are interesting; however, it is more common to formulate specific hypotheses in this kind of literature. I think the formulation of specific hypotheses will make the goals of the study clearer.

Thank you for the suggestion, we reformulated the aim to hypotheses. 

3. The methodology is very well explained and complete. However, there are some details that I think can improve the paper. Firstly, the pre-test sample is not specified; is it the same as the main study? Was the pre-test done in the same session of the main study or before? Secondly, it is unclear who the main study sample is; are they students, or were the data collected through an online platform? If the authors used an online platform (e.g., Mturk or Prolific), how did you assure people receiving one of the products?

Thank you addressing this unclarity. We specified that the pretest was performed in a separate session prior to the main study. 

The participants of the main study were recruited by convenience sampling and hence consisted of people of different age groups and occupations. We randomly drew 3 participants from all participants who indicated their e-mail address to take part in the lottery and contacted them to deliver their chosen product.

4. The results are interesting and well presented. I have only a few comments. What is the statistic that the authors used to compare the strength of the attraction effect in healthy product conditions and unhealthy product conditions (line 189)? The authors can conclude that one is significant and the other just marginal; however, it is not clear how the authors can conclude that one is stronger than the other one. Moreover, when performing a binary logistic regression is more complete to report also the beta coefficients and the R2, while in the analysis there is only a chi-square. It is also not clear if the authors reported there (e.g., line 181) the regression results where they controlled for self-control, diet, gender, and age or the one without those variables. Maybe the regression table could also be reported in the results section since the paper is not too long.

Thank you for pointing out the ambiguous description of the strength of the attraction effect in the healthy vs. unhealthy choice set. We indeed did not directly compare the difference in strength between the two choice set; we tested the presence of the attraction effects in the two choice sets separately. We adjusted the text accordingly.

We added the beta coefficients and R2 in the binary logistics regression reports; thank you for your comment. 

The regression results we report include the control variables self-control, diet, gender and age, as indicated in line 191 with track changes and 186 without track changes. For clarification, we now also indicated this in lines 195-196 with track changes and 190-191 without track changes. 

5. Finally the discussion is well written and well motivated.

In general, I think the manuscript makes a valuable contribution, it is well written, and if the few reported details were addressed, I would suggest acceptance.

Reviewer #2: I though this paper is well written, and information is presented in a concise and relevant way, which I appreciate. The novelty of this study comes from the novel experiment design which allows exploring effects of attractions across new product mix sets.

The results are interesting from the practical point of view, although this is just a first step as validating the findings beyond an online experiment in a practical setting and for various product types is needed. Since this study is like a gateway towards more realistic and more product inclusive further studies, I believe the concluding section could benefit from more extensive coverage of what needs to be done in the future and where these types of results may have best use. I think that my suggestion in 1) below should take care of this.

I would suggest few revisions to improve the overall context and the flow to the reader.

1) I think the limitations of this study, which are summarized in one short paragraph (lines 238-241) should be expanded and communicated more formally. I would think that participant’s demographics may have an impact on the results of this experiment. For example, current sample was mostly females, perhaps the results are biased towards this group, thus gender, as well as generational attributes (age of the participants) may also be contributing factors in the results found in this study.

We thank you for this insightful comment and reformulated and expanded the limitations accordingly.

2) It may be wise to include in the discussion of an ongoing Covid-19 Pandemic, as it may be also proving to have severe negative emotional impact on people’s eating habits (well documented in research studies by now). Depending on extend of the relevance to the current study, this could be accomplished in the introduction section or discussion. The study does mention repeatedly the importance of the environment, thus pandemic impact may be linked to that.

We thank you for this suggestion. We included the practical relevance of our study (that ran prior to Covid-19) to the current pandemic in the discussion. 

3) I would like to see illustrative example with an apple and muffin (line 77) to be complete with an example of a decoy product choice appropriate for such a product mix. It reads unfinished.

We rephrased the example to include the decoy immediately. Thank you for the suggestion. 

4) Line 185: change significant to significantly

Thank you for noticing this, we adjusted accordingly.

Several references might be useful here:

Brody, L. R., Hall, J. A. & Stokes, L. R. (2018). Gender and emotion: Theory, findings, and context. In L. F. Barrett, M. Lewis & J. M. Haviland-Jones (Eds.), Handbook of emotions (4th ed., pp. 369– 392). 

Bove, C. F., Sobal, J. & Rauschenbach, B. S. (2003). Food choices among newly married couples: Convergence, conflict, individualism, and projects. Appetite, 40, 25– 41.

Marty, L., de Lauzon-Guillain, B., Labesse, M. & Nicklaus, S. (2021). Food choice motives and the nutritional quality of diet during the COVID-19 lockdown in France. Appetite, 157, 105005. https://doi.org/10.1016/j.appet.2020.105005

Nazzaro, C., Lerro, M. & Marotta, G. (2018). Assessing parental traits affecting children’s food habits: An analysis of the determinants of responsible consumption. Agricultural and Food Economics, 6, 23.

Poelman, M. P., Gillebaart, M., Schlinkert, C., Dijkstra, S. C., Derksen, E., Mensink, F. et al. (2021). Eating behavior and food purchases during the COVID-19 lockdown: A cross-sectional study among adults in the Netherlands. Appetite, 157, 105002. https://doi.org/10.1016/j.appet.2020.105002.

Mahmoud, A. B., Hack-Polay, D., Fuxman, L., & Nicoletti, M. (2021). The Janus-faced effects of COVID-19 perceptions on family healthy eating behavior: Parent’s negative experience as a mediator and gender as a moderator. Scandinavian journal of psychology. https://doi.org/10.1111/SJOP.12742

We are grateful for these interesting suggested references. We had a careful look at all of them and included the references we deemed relevant.

---

## [Decision Letter · Decision Letter 1]

21 Oct 2021

Attract the best: The attraction effect as an effective strategy to enhance healthy choices

PONE-D-21-15713R1

Dear Dr. van den Enden,

We’re pleased to inform you that your manuscript has been judged scientifically suitable for publication and will be formally accepted for publication once it meets all outstanding technical requirements.

Kind regards,

Ali B. Mahmoud, Ph.D.

Academic Editor

PLOS ONE

Additional Editor Comments (optional):

Reviewers' comments:

Reviewer's Responses to Questions

**Comments to the Author**

1. If the authors have adequately addressed your comments raised in a previous round of review and you feel that this manuscript is now acceptable for publication, you may indicate that here to bypass the “Comments to the Author” section, enter your conflict of interest statement in the “Confidential to Editor” section, and submit your "Accept" recommendation.

Reviewer #1: All comments have been addressed

Reviewer #2: All comments have been addressed

2. Is the manuscript technically sound, and do the data support the conclusions?

Reviewer #1: Yes

Reviewer #2: Yes

3. Has the statistical analysis been performed appropriately and rigorously? 

Reviewer #1: Yes

Reviewer #2: Yes

4. Have the authors made all data underlying the findings in their manuscript fully available?

Reviewer #1: Yes

Reviewer #2: Yes

5. Is the manuscript presented in an intelligible fashion and written in standard English?

Reviewer #1: Yes

Reviewer #2: Yes

6. Review Comments to the Author

Reviewer #1: The authors successfully addressed the comment I made on the first round of reviews, therefore I suggest acceptance.

However, I would still suggest that the regression table be reported in the main text and for the characteristics of the pre-test sample to be written in the methodology section.

Reviewer #2: Thank you for addressing all comments in an efficient fashion. Please reconcile font size changes: Line 111 & Line 204-205

7. PLOS authors have the option to publish the peer review history of their article (what does this mean?). If published, this will include your full peer review and any attached files.

Reviewer #1: No

Reviewer #2: No

---

## [Editor Report · Acceptance letter]

27 Oct 2021

PONE-D-21-15713R1 

Attract the best: The attraction effect as an effective strategy to enhance healthy choices 

Dear Dr. van den Enden:

I'm pleased to inform you that your manuscript has been deemed suitable for publication in PLOS ONE. Congratulations! Your manuscript is now with our production department. 

Kind regards, 

on behalf of

Dr. Ali B. Mahmoud 

Academic Editor

PLOS ONE